# The Omicron Variant Reinfection Risk among Individuals with a Previous SARS-CoV-2 Infection within One Year in Shanghai, China: A Cross-Sectional Study

**DOI:** 10.3390/vaccines11071146

**Published:** 2023-06-25

**Authors:** Chuchu Ye, Ge Zhang, Anran Zhang, Hualei Xin, Kang Wu, Zhongjie Li, Yilin Jia, Lipeng Hao, Caoyi Xue, Yuanping Wang, Hongmei Xu, Weiping Zhu, Yixin Zhou

**Affiliations:** 1Shanghai Pudong New Area Center for Disease Control and Prevention, Shanghai 200136, China; cynthia-cloth@163.com (C.Y.); cdczhanganran@126.com (A.Z.); kwu@pdcdc.sh.cn (K.W.); yljia@pdcdc.sh.cn (Y.J.); hlpmail@126.com (L.H.); cyxue@pdcdc.sh.cn (C.X.); ypwang@pdcdc.sh.cn (Y.W.); hmxu@pdcdc.sh.cn (H.X.); 2School of Public Health, Dali University, Dali 671003, China; 18735110959@163.com; 3World Health Organization Collaborating Centre for Infectious Disease Epidemiology and Control, School of Public Health, Li Ka Shing Faculty of Medicine, The University of Hong Kong, Hong Kong SAR, China; u3007495@connect.hku.hk; 4School of Population Medicine and Public Health, Chinese Academy of Medical Sciences and Peking Union Medical College, Beijing 100073, China; lizhongjiecdc@163.com

**Keywords:** reinfection, COVID-19, vaccination, Omicron

## Abstract

Reinfection with severe acute respiratory syndrome coronavirus 2 (SARS-CoV-2) variants due to immune escape is challenging for the global response to the pandemic. We estimated the Omicron reinfection prevalence among people who had a previous SARS-CoV-2 infection in Shanghai, China. We conducted a telephone survey in December 2022 with those who had previously been infected with Omicron between March and May 2022. Information on their demographics, coronavirus disease 2019 (COVID-19) testing, and vaccination history was collected. The overall and subgroup reinfection rates were estimated and compared. Among the 1981 respondents who were infected between March and May 2022, 260 had positive nucleic acid or rapid antigen tests in December 2022, with an estimated reinfection rate of 13.1% (95% confidence interval [95% CI]: 11.6–14.6). The reinfection rate for those who had a booster vaccination was 11.4% (95% CI: 9.2–13.7), which was significantly lower than that for those with an incomplete vaccination series (15.2%, 95% CI: 12.3–18.1) (adjusted odds ratio [aOR]: 0.579; 95% CI: 0.412–0.813). Reinfection with the Omicron variant was lower among individuals with a previous SARS-CoV-2 infection and those who had a booster vaccination, suggesting that hybrid immunity may offer protection against reinfection with Omicron sublineages.

## 1. Introduction

In recent years, the emergence of severe acute respiratory syndrome coronavirus 2 (SARS-CoV-2) has resulted in a significant worldwide public health crisis, profoundly impacting human physical and mental well-being, the global economy, and sociopolitical landscapes. To date, there have been over 754 million confirmed cases of SARS-CoV-2 globally, with 6.82 million reported deaths as of February 2023 [1]. Since the emergence of SARS-CoV-2 in December 2019, multiple variants of concern have arisen and rapidly disseminated globally [2]. The initial predominant Omicron variant of SARS-CoV-2, BA.1, contains 35 mutations in its spike protein compared to the original variant that emerged in late 2019 [3]. Shortly after its identification, the BA.1 variant quickly emerged as the prevailing variant on a global scale and has subsequently undergone further genetic changes, giving rise to multiple sublineages. On 26 November 2021, the World Health Organization (WHO) designated B.1.1.529 as a variant of concern based on recommendations from the WHO Technical Advisory Group on Virus Evolution [4].

An essential aspect of any infectious disease is to determine whether its infection results in long-lasting immunity or whether recurrent reinfection is prevalent. Both natural immunity acquired from a prior infection and vaccine-induced immunity against COVID-19 play crucial roles in reducing the severity and impact of this disease. However, several knowledge gaps remain concerning the risk of reinfection following previous exposure to different variants of SARS-CoV-2 [5,6,7]. During the initial waves of SARS-CoV-2, including the wild-type, alpha, and delta variants, the prevailing belief was that infection conferred long-lasting immunity. However, more recent evidence, especially during the Omicron waves in 2022, has indicated that reinfection can occur relatively frequently [8,9,10,11,12].

During the first year of the COVID-19 pandemic, certain countries, including China, Singapore, Australia, and New Zealand, implemented strategies to effectively suppress community transmission and successfully maintained containment measures [13,14]. Prior to December 2022, China implemented a “zero COVID” policy, which aimed to achieve and maintain zero tolerance for the local transmission of SARS-CoV-2. This policy focused on sustained containment by effectively preventing and responding to any externally introduced outbreaks. In the event of an outbreak, the response measures were based on an assessment of the epidemic risk and utilized the same strategies employed during the initial containment phase [15]. These measures were further reinforced by stringent border protection measures to minimize the occurrence of imported outbreaks. Additionally, routine polymerase chain reaction (PCR) testing was extensively conducted to enable a highly sensitive surveillance for detecting infection [16,17,18].

Shanghai is the largest city in eastern China. According to Next Generation Sequencing (NGS) technology, which was applied to detect the whole genome of SARS-CoV-2 for each COVID-19 outbreak, Shanghai experienced the first wave of COVID-19 caused by the Omicron BA.2 variant [19,20] between March and May 2022 and the second wave caused by the circulating Omicron BA5.2 and BF.7 variants in December 2022, right after the downgrade of the “zero COVID” policy. A key question with the emergence of these new variants is the extent to which they are able to reinfect those who have had a prior natural infection. This reinfection rate cannot be ascertained without routine PCR testing in the community.

This study aimed to assess the reinfection risk among people with confirmed COVID-19 during the 2022 spring outbreak within one year and to explore the effect of hybrid immunity (i.e., vaccination vs. non-vaccination) on reinfection. The findings of this study will provide scientific evidence for the implementation of appropriate intervention strategies and programs for targeting the oncoming waves of COVID-19 in Shanghai and other areas, with the possibility of considering a subsidy policy for COVID-19 vaccination.

## 2. Materials and Methods

### 2.1. Study Design

A cross-sectional survey was conducted to assess the prevalence of reinfection among previously infected individuals during the second wave of the outbreak in December 2022.

Since the strict quarantine and frequent screening policy for COVID-19 management was implemented before 1 December 2022, in China, we assumed that all the potential cases would be identified during the first wave in spring of 2022. From 1 March to 31 May 2022, there were 245,803 new nucleic-acid-positive cases in Pudong New Area, according to the local nucleic acid testing information system. With the downgrade of the disease control policy after 1 December 2022, a great portion of these cases would not be identified because frequent PCR testing had stopped. Nonetheless, we still found that 5649 of the 245,803 previously infected individuals tested positive again according to the same system in December 2022. We estimated that the lowest reinfection rate was 2.30% (5649/245,803).

We conducted a stratified sampling method in this study. The estimated response rate was 60% according to the pilot study. A sample size of 3361 participants was determined for this cross-sectional study based on various factors. These factors included an estimated reinfection rate of 2.30%, an alpha risk of 5%, a maximum permissible error of 0.01, and a design effect of 2. The minimum sample size was calculated using the following formula commonly used in cross-sectional studies: N = 2(Z/δ)2p(1 − p).

Telephone interviews were conducted from January 17 to 31, 2023, in Pudong New Area, Shanghai, eastern China. The participants included only permanent residents who had lived in Pudong New Area for ≥12 months. The guardians of children aged 6 months to 14 years were interviewed. The data collection was conducted by professional investigators at the Shanghai Pudong New Area Center for Disease Control and Prevention (PDCDC). Each selected respondent participated in the study and provided their responses to a questionnaire. The interviewers explained the questionnaire items to the respondents and CDC professionals recorded their answers in a standardized questionnaire.

For the purpose of this study, reinfection was defined as a positive result using a polymerase chain reaction (PCR) or rapid antigen test conducted between 1 December and 31 December 2022.

### 2.2. Data Sources and Description

The questionnaire used in the telephone survey included three sets of questions regarding the following: (i) sociodemographic variables, including age and sex; (ii) the occurrence of SARS-CoV-2 reinfection, as defined by a positive PCR or rapid antigen test; and (iii) COVID-19 vaccination status before the survey. The investigators asked the participants about their nucleic acid/antigen test results and the time of this testing. If the participants had not experienced reinfection, the other questions regarding reinfection were not asked. For those who declined to answer any part of the questionnaire, the remaining sections of the questionnaire were not completed. According to the type and dose of the vaccine, we divided the respondents’ vaccination statuses into an incomplete vaccination series (unvaccinated or had one inactivated vaccine 14 days or longer before 1 December 2022), a complete vaccination series (having received two doses of an inactivated vaccine 14 days or longer before 1 December 2022), or a booster vaccination series (having received three or more doses of an inactivated vaccine 14 days or longer before 1 December 2022).

The questionnaire was validated in a pilot survey conducted in a small area before the formal survey.

### 2.3. Statistical Analysis

The proportions of individuals who experienced SARS-CoV-2 reinfection were calculated by dividing the number of reinfections by the total number of respondents. These proportions were then stratified by sex and age group. Pearson’s chi-square test was employed to compare the reinfection risks across the different subgroups.

A multivariable logistic regression analysis was conducted to identify the potential factors influencing this risk of reinfection. Adjusted odds ratios (ORs) with corresponding 95% confidence intervals (CIs) were calculated to examine the associations between these factors and the likelihood of reinfection.

A two-sided *p* value less than 0.05 was considered to be statistically significant. All the analyses were conducted in R version 4.4.2 (R Core Team, R: A language and environment for statistical computing. R Foundation for Statistical Computing, Vienna, Austria).

## 3. Results

### 3.1. Basic Characteristics of Respondents

Among the 3361 respondents, 522 (15.53%) were excluded because they were not permanent residents in the study area, 474 (14.10%) could not be contacted, 36 (1.07%) had died before 1 December, and 348 (10.36%) refused to participate (Figure 1). A total of 1981 valid questionnaires were finally collected, yielding a response rate of 58.94% (1981/3361) (Table 1).

The study population consisted of respondents with ages ranging from 0.9 to 99.8 years, with a median age of 45.3 (interquartile range (IQR): 32.8–57.1) years. Among the respondents, women accounted for 43.7% (866/1981) of the total sample. The majority of respondents, 59.6%, were aged 30–59 years; respondents younger than 20 years and older than 70 years accounted for 7.1% and 9.8%, respectively. The proportions of the respondents who received an incomplete vaccination series, a complete vaccination series, and a booster vaccination series were 29.2%, 31.9%, and 38.9%, respectively (Table 2).

### 3.2. Reinfection Rate of SARS-CoV-2 among Different Populations

A total of 260 respondents reported reinfection during the December 2022 outbreak. This reinfection risk was estimated to be 13.12% [95% CI: 11.64–14.61] (Table 2).

The reinfection risk among the male respondents (14.69%, 95% CI: 12.59–16.79) was significantly higher than that among the female respondents (11.19%, 95% CI: 9.11–13.26). Among the different age groups, adults aged 30–39 years had the highest reinfection risk (21.07%, 95% CI: 17.13–25.00) and children younger than 9 years had the lowest reinfection risk (2.74%, 95% CI: 0–6.48).

A total of 1501 respondents received at least one dose of a SARS-CoV-2 vaccine, with a coverage rate of 75.77%. The reinfection risks for people who received an incomplete vaccination series, complete vaccination series, and booster vaccination series were 15.20% (95% CI: 12.27–18.12), 13.29% (95% CI: 10.64–15.94), and 11.43% (95% CI: 9.18–13.68), respectively.

Among the respondents without reinfection, women constituted 45.7% (786/1721) of the total sample. Within this group, 58.3% were aged 30–59 years. The proportions of respondents without reinfection who received an incomplete vaccination series, complete vaccination series, and booster vaccination series were 28.5%, 31.8%, and 39.6%, respectively.

The first reinfection case was detected on 2 December. The epidemic curve increased sharply after 12 December and peaked on 20 December. The time series of SARS-CoV-2 reinfection for individuals who had a history of previous infection is depicted in Figure 2.

### 3.3. Factors That Influenced Reinfection

As shown in the logistic regression model (Table 3), female sex (aOR = 0.732, 95% CI: 0.557–0.961, *p* = 0.0245), an age younger than 9 years (aOR = 0.163, 95% CI: 0.035–0.764, *p* = 0.0213), and a booster vaccination series (aOR = 0.579, 95% CI: 0.412–0.813, *p* = 0.0016) were significantly associated with a decreased risk of reinfection.

## 4. Discussion

### 4.1. Main Findings

Our study found that the reinfection rate of the Omicron variant among people who had a previous SARS-CoV-2 infection within one year was 13.12%. Moreover, a cohort study was conducted among people from the community in the study area, which included over 2500 noninfected individuals, with a vaccine coverage of 70.3%. As of December 2022, the crude infection rate was as high as 75% in this cohort, over five times higher than that among previously infected people. A natural BA.2 infection provided strong protection against infection during the Omicron outbreak caused by BA5.2 or BF.7. The reinfection rates among females, children younger than 9 years, and people with a booster vaccination history were significantly lower than those among other groups.

### 4.2. Reinfection Rate of Omicron

A natural infection with SARS-CoV-2 elicits strong protection against reinfection with the B.1.1.7 (Alpha), B.1.351 (Beta), and B.1.617.2 (Delta) variants. However, the B.1.1.529 (Omicron) variant harbors multiple mutations that can mediate immune evasion [8]. A study in Turkey showed that reinfection occurred with 520 (13.0%) of 3992 Omicron sublineages, which is similar to the findings of our study [11].

Recent studies have found that the risk of reinfection is higher for Omicron than for the other strains of SARS-CoV-2 [21,22,23]. According to a meta-analysis from the University of Ferrara in Italy, the reinfection rate of SARS-CoV-2 gradually increased, with rates of 0.57% for Alpha, 1.25% for Delta, and 3.31% for Omicron; that for Omicron was 5.8 times higher than that for Alpha [24]. Another study found that this reinfection rate was only 0.7% among people who were infected for the first time by Omicron BA.4/BA.5. However, if an infected person was first infected with the Delta or Omicron BA.2 strain, then their chances of being reinfected with Omicron BA.4/BA.5 were greater [25].

One study indicated that the risk of reinfection increased almost 18-fold following the emergence of the Omicron variant compared to that of the Delta variant [26]. Moreover, compared to Alpha and Delta, the decrease in antibody protection was greater after an infection with Omicron. The effectiveness of the antibodies in infected individuals at 3 to 5 months after infections with Alpha and Delta still reached 86.6% and 91.3%, respectively. The decrease rate was limited among people infected with Omicron, but the lowest value was still above 60% [25]. A recent study found that protection conferred by a previous Omicron infection was moderate, at approximately 50% when the previous infection was with a BA.1 or BA.2 subvariant, but was approximately 80% when the previous infection was with a BA.4 or BA.5 subvariant [27].

### 4.3. Reinfection Rates among Different Groups

Overall, the reinfection rates across the various age groups were generally comparable, except for the younger population, for which the rates were notably lower than those observed for the other age groups. A population-level retrospective cohort study conducted in Kuwait from 2020 to 2021 also found that SARS-CoV-2 reinfection was uncommon among children [28]. One study conducted in France between March 2021 and February 2022 found that people younger than 18 years and older than 40 years had lower rates of reinfection (*p* < 0.001) [22].

A retrospective epidemiological study analyzed the SARS-CoV-2 reinfection cases in Bahrain between 1 April 2020 and 23 July 2021, using data from the Bahrain national COVID-19 database of individuals who had two positive test results for SARS-CoV-2 at least 3 months apart. The researchers found that a significantly larger proportion of reinfected individuals were male (60.3%, *p* < 0.0001) and that these reinfection episodes were highest among those aged 30–39 years (29.7%) [29], which is consistent with our study.

We conducted this study in January 2023, when the second wave of the Omicron variant in China had not completely ended. It has been shown that the interval between two infection events varies by time, strain, vaccine situation, and other factors [2,6,30,31]. The protection conferred by a prior infection against reinfection with pre-Omicron variants was initially high and maintained a consistently high level, even after 40 weeks [12]. A study using the whole-genome viral RNA sequencing of clinical specimens collected during the initial infections and suspected reinfections of four healthcare workers at the Habib Bourguiba University Hospital, who retested positive for SARS-CoV-2 via an RT—PCR after recovery, showed a range between 45 and 141 days [32]. The results of a retrospective, longitudinal analysis among healthcare workers suggested that the first episode of a SARS-CoV-2 infection provides strong protection against reinfection and that this lasts for at least a year, including during periods of high transmission in the community. At a median follow-up of 38.4 (range: 7.1–55.0) weeks following this initial infection, the cumulative actuarial probability of a SARS-CoV-2 infection at 52 weeks was determined to be 2.2% (95% CI, 1.0–4.9%) [33]. Another study found that the fewest reinfection episodes occurred 3–6 months after the first infection, with most occurring ≥9 months after the initial infection [29].

### 4.4. Hybrid Immunity against Omicron

Hybrid immunity, particularly against the Omicron variant, has been widely recognized as the most resilient approach to combating SARS-CoV-2 [34,35,36]. Previous research has demonstrated that a combination of naturally acquired immunity through multiple reinfections and vaccine-induced immunity confers significant protection against severe SARS-CoV-2 disease and mortality [20,37,38,39]. Irrespective of the prevailing virus variant, the most significant risk factor for reinfection was found to be a lack of vaccination [26,40].

In Shanghai, vaccines against SARS-CoV-2 were first offered in the end of 2020. Of the total respondents, 75.77% (1501/1981) had received vaccination with at least one dose. Booster-vaccinated individuals had the lowest reinfection rate, followed by those who were fully vaccinated. The highest risk of reinfection was observed among unvaccinated or incompletely vaccinated people. Evidence for triple-vaccinated individuals has shown that a previous Omicron infection provides high amounts of protection against BA.5 and BA.2 infections [30]. Furthermore, a post-Omicron infection with a booster vaccination provides high protection against BA.4, BA.5, and XBB reinfection; however, the protection against XBB is much lower compared to BA.4 and BA.5 [41]. A study on the incremental protection and durability of infection-acquired immunity against Omicron infection among individuals with hybrid immunity in Canada showed that a previous SARS-CoV-2 infection provided added cross-variant immunity with vaccination [42]. According to a population-level observational study, unvaccinated, incompletely, or completely vaccinated individuals were slightly more likely to be reinfected than those who were recipients of a third (booster) vaccine dose [43]. The estimated protection (95% CI) against Omicron infection was found to be consistently significantly higher among vaccinated individuals with a prior infection compared to vaccinated infection-naive individuals, with 65% vs. 20% for one dose, 68% vs. 42% for two doses, and 83% vs. 73% for three doses [21]. However, we also noticed that the effectiveness of booster vaccination might provide a short period of protection [44].

In our study, we reported, for the first time, the reinfection rates during the first Omicron outbreak after the downgrade of China’s “zero COVID” policy among people who had been infected in the 2022 spring Omicron outbreak in Shanghai. In addition, we provided evidence for the effectiveness of inactivated vaccination, such as Sinopharm and CoronaVac, on the prevention of BA.5.2 and BF.7 reinfections among people with a previous post-Omicron infection. With the ongoing emergence of SARS-CoV-2 variants becoming a global concern, immunity plays a crucial role in combating SARS-CoV-2 and its variants. Protective immunity derived from immune memory serves as a defense mechanism against SARS-CoV-2. Recent studies have highlighted the robustness of hybrid immunity, which combines naturally acquired and vaccine-induced immunity, in providing the highest level of protection against the virus. Our study indicates that protection is strong after natural immunity within one year against the different Omicron variant sublineages. Furthermore, evidence of the effectiveness of hybrid immunity was also found, consistent with other studies conducted worldwide.

### 4.5. Limitations

This study shares the limitations commonly associated with retrospective surveys and cross-sectional study designs, which include the potential for recall bias and selection bias. Before December 2022, routine PCR testing was performed in the community, and hypothetically, all potential cases would be identified and registered. The sample of our study was obtained from the local PCR registration system. However, many cases involved migrants, travelers, or floating workers who stayed in Shanghai during the 2022 spring outbreak. They had left the city when our survey was conducted after the downgrade of the prevention policy. This was the major reason for the low response rate, especially among young males. In addition, after this downgrade of the prevention policy, only a small number of individuals with suspected disease sought PCR testing or rapid antigen testing. Obviously, asymptomatic individuals and younger or older adults for whom PCR or rapid antigen testing was less convenient had limited opportunities for confirming a reinfection, and such people may have provided negative answers to the survey. This bias might also be one of the reasons for the lower reinfection rates among younger children and older people. Additionally, the rapid antigen testing was performed by the respondents themselves, so there was still bias associated with the diagnostic tests, which might have led to an underestimation of the reinfection rate, since some infected respondents might not have been detected as positive due to their ability for sampling and testing.

## 5. Conclusions

To summarize, our study findings indicated that individuals with a previous SARS-CoV-2 infection within one year had a significantly lower risk of reinfection than the general population. Booster vaccination was significant factor in reducing this reinfection risk.

## Figures and Tables

**Figure 1 vaccines-11-01146-f001:**
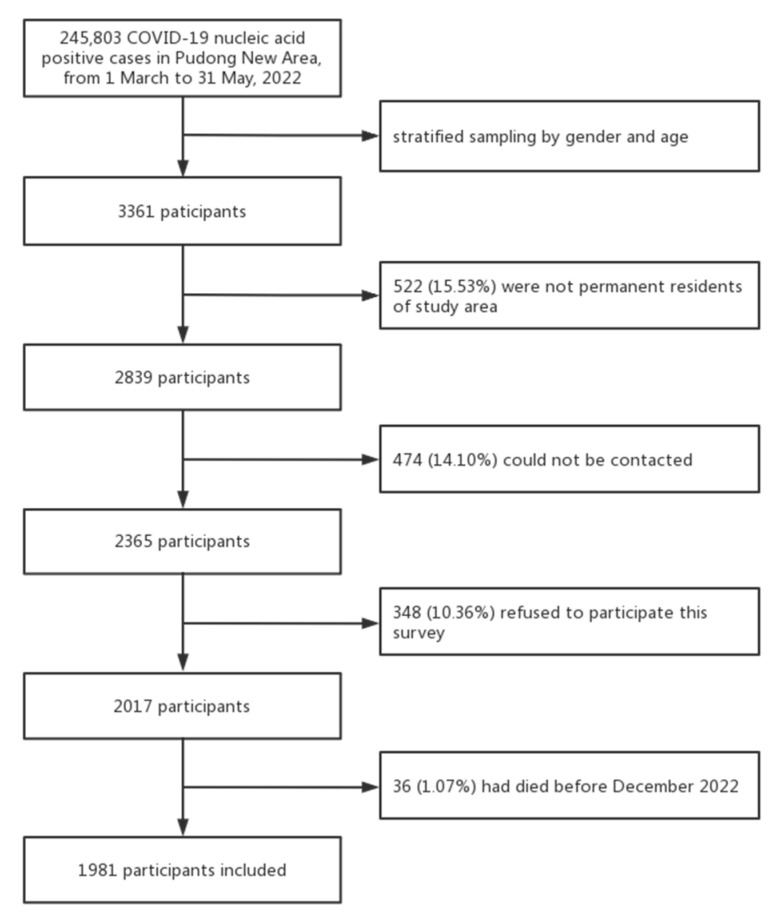
Flowchart of participant selection.

**Figure 2 vaccines-11-01146-f002:**
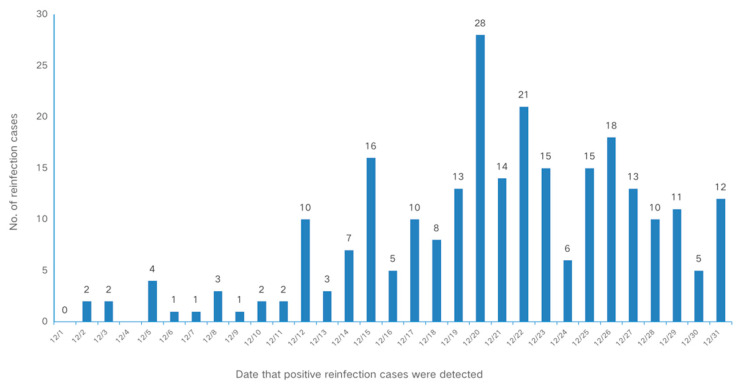
Epidemic curve of 260 reinfection cases in Shanghai, China, 2022.

**Table 1 vaccines-11-01146-t001:** Characteristics of the study participants.

Sex	Age Group	COVID-19 Infections from March to May 2022 N = 245,803	No. of People with COVID-19 Sampled N = 3361	No. of People with COVID-19 Who Responded N = 1981	Response Rate (%)
Male	0–9	4453 (1.81%)	53 (1.58%)	40 (2.02%)	65.57
	10–19	5653 (2.3%)	76 (2.26%)	38 (1.92%)	49.35
	20–29	22,151 (9.01%)	333 (9.91%)	153 (7.72%)	50.50
	30–39	30,411 (12.37%)	405 (12.05%)	238 (12.01%)	57.21
	40–49	24,528 (9.98%)	365 (10.86%)	199 (10.05%)	59.40
	50–59	28,536 (11.61%)	376 (11.19%)	229 (11.56%)	58.72
	60–69	14,337 (5.83%)	189 (5.62%)	106 (5.35%)	54.08
	70–79	7038 (2.86%)	96 (2.86%)	59 (2.98%)	61.46
	80+	3310 (1.35%)	55 (1.64%)	34 (1.72%)	75.56
Female	0–9	3826 (1.56%)	49 (1.46%)	33 (1.67%)	63.46
	10–19	3946 (1.61%)	53 (1.58%)	29 (1.46%)	53.70
	20–29	12,485 (5.08%)	182 (5.42%)	95 (4.80%)	55.56
	30–39	20,122 (8.19%)	244 (7.26%)	175 (8.83%)	63.64
	40–49	18,182 (7.4%)	232 (6.90%)	165 (8.33%)	66.27
	50–59	20,647 (8.4%)	300 (8.93%)	174 (8.78%)	61.70
	60–69	13,721 (5.58%)	170 (5.06%)	113 (5.70%)	60.11
	70–79	7245 (2.95%)	103 (3.06%)	60 (3.03%)	60.61
	80+	5212 (2.12%)	80 (2.38%)	41 (2.07%)	57.75

**Table 2 vaccines-11-01146-t002:** SARS-CoV-2 reinfection rate among individuals with past infection in the December 2022 outbreak, Pudong New Area, Shanghai (n = 1981).

Characteristics	No. of Respondents	Proportions (%)	No. of Reinfections	Adjusted Reinfection Rate (95% CI)	*p*
Total	1981	100.0	260	13.12 (11.64–14.61)	
Sex					
Male	1115	56.3	161	14.69 (12.59–16.79)	0.025
Female	866	43.7	99	11.19 (9.11–13.26)	
Age					
0–9	73	3.7	2	2.74 (0–6.48)	0.000
10–19	67	3.4	6	8.96 (2.12–15.79)	
20–29	248	12.5	33	13.31 (9.08–17.53)	
30–39	413	20.8	87	21.07 (17.13–25.00)	
40–49	364	18.4	47	12.91 (9.47–16.36)	
50–59	403	20.3	42	10.42 (7.44–13.41)	
60–69	219	11.1	22	10.05 (6.06–14.03)	
70–79	119	6.0	10	8.40 (3.42–13.39)	
80+	75	3.8	11	14.67 (6.66–22.67)	
Vaccination					
Incomplete	579	29.2	88	15.20 (12.27–18.12)	0.022
Complete	632	31.9	84	13.29 (10.64–15.94)	
Booster	770	38.9	88	11.43 (9.18–13.68)	

**Table 3 vaccines-11-01146-t003:** Factors that influenced the SARS-CoV-2 reinfection rate among individuals with past infection in the December 2022 outbreak, Pudong New Area, Shanghai (n = 1981).

Characteristics	No. of Reinfections	Adjusted Odds Ratio (95% CI)	*p*
Total	260		
Sex			
Male	161	REF	0.0245
Female	99	0.732 (0.557–0.961)	
Age			
0–9	2	0.163 (0.035–0.764)	0.0213
10–19	6	0.664 (0.226–1.949)	0.4565
20–29	33	1.086 (0.509–2.317)	0.8316
30–39	87	2.034 (0.997–4.148)	0.0510
40–49	47	1.156 (0.551–2.425)	0.7006
50–59	42	0.903 (0.429–1.903)	0.7888
60–69	22	0.823 (0.371–1.823)	0.6305
70–79	10	0.643 (0.256–1.616)	0.3477
80+	11	REF	
Vaccination			
Incomplete	88	REF	
Complete	84	0.741 (0.528–1.042)	0.0849
Booster	88	0.579 (0.412–0.813)	0.0016

## Data Availability

The data that support the findings of this study are available from the corresponding author, Y.Z., upon reasonable request.

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
