# Peer review of "The Omicron Variant Reinfection Risk among Individuals with a Previous SARS-CoV-2 Infection within One Year in Shanghai, China: A Cross-Sectional Study"

_vaccines, 2023, doi:10.3390/vaccines11071146_

Round 1
Reviewer 1 Report
I reviewed the manuscript titled ' Omicron variant reinfection risk among individuals with previous SARS-CoV-2 infections within one year in Shanghai, China: A cross-sectional study'.
This is an interesting study, however, some concerns need to be addressed:
1- There are some similar studies on the COVID reinfection risk among past infected patients. The novelty of the present study should be explained. Moreover, past studies should be discussed better.
2- How did the authors detect that re-infection was due to Omicron in all patients, not possibly the other species?
3- How can the authors prove that past COVID history had been a protective factor for re-infection? and not COVID vaccination?
4- It was better to run a similar analysis on the group without re-infection and compare.
5- There are serious bias sources that made conclusions to be written accordingly with catios.
English sounds appropriate, however, the authors encouraged to revise it during scientific editions.
Author Response
We thank for the reviewer's suggestions and have revised our manuscript according to your advice.

Reviewer 2 Report
The manuscript is perfectly designed, it ´s clear and well documented and explained.
I have made some comments about possible bias (see attached document

Author Response
- Line 116: I suppose that sampling and handling for PCR was done by any processional working for specific laboratories while for rapid antigen test could be done by yourself, is it correct? in that case, could it introduce any bias? because the ability to take the sample is not the same. I suggest to comment it in that point
Response: We thank for the reviewer’s advice and agree with your opinion. Rapid antigen test was done by the respondents themselves, there was still bias associated to the diagnostic tests, which might led to the underestimation of the reinfection rate, since some infected respondents might not be detected positive due to their ability of sampling and testing.
We have added this statement in the discussion part.
- Line 169:only 2 reinfected cases, significant?
Response: We thank for the reviewer’s remind and checked our original data. Only 2 children were reinfected among all the 40 cases, however, in the Chi-squre table, theoretical frequency in this group was over 5 and we believe the statistical method was appropriate in this part.
- Table 3:I suggest to remark the significant associated factors (in bold).
Response: We thank for the reviewer’s suggestion and have made related revision in our manuscript.
- Line 310 :in my opinion, potential bias associated to the diagnostic tests (sampling and handling efficiency based on people doing it) must be included
Response: We thank for the reviewer’s remind and agree with your opinion. As response to your first comment, rapid antigen test was done by the respondents themselves, there was still bias associated to the diagnostic tests, which might led to the underestimation of the reinfection rate, since some infected respondents might not be detected positive due to their ability of sampling and testing.
We have added this statement in the discussion part.

Reviewer 3 Report
· The method of reporting the results and the method of statistical analysis of the data and the statistical tests used (fully) should be mentioned. Also, it is necessary to check the presuppositions of the statistical tests used and report their implementation (by reporting the relevant figures).
· in the work method; Was it random in collecting the samples? Sample size formula
· The discussion needs to be fundamentally rewritten:
· The results of other studies, whether similar or different, should be brought and compared and inferred, and what is the difference between the current article and those articles?
· Mention each of your important findings, then compare with others and mention your interpretation and inference of the results and causes of similarities and differences.
· Indicate the application of the results
· The limitations of the study should be mentioned
· running title should be added
· -The type of study should be mentioned.
· -Abbreviations should have been explained at least once
-It is better for the authors to carefully edited by a fluent native English
Round 2
Reviewer 1 Report
My comments have been addressed well by the authors.